# Multiple antagonist calcium-dependent mechanisms control CaM kinase-1 subcellular localization in a *C. elegans* thermal nociceptor

**Domenica Ippolito, Dominique A Glauser***

Department of Biology, University of Fribourg, Fribourg, Switzerland

**Abstract** Nociceptive habituation is a conserved process through which pain sensitivity threshold is adjusted based on past sensory experience and which may be dysregulated in human chronic pain conditions. Noxious heat habituation in *Caenorhabditis elegans* involves the nuclear translocation of CaM kinase-1 (CMK-1) in the FLP thermo-nociceptors neurons, causing reduced animal heat sensitivity and avoidance responses. The phosphorylation of CMK-1 on T179 by CaM kinase kinase-1 (CKK-1) is required for nuclear entry. Recently, we identified a specific nuclear export sequence (NES) required to maintain CMK-1 in the cytoplasm at rest (20°C) and showed that $Ca^{2+}$/CaM binding is sufficient to enhance CMK-1 affinity for IMA-3 via a specific nuclear localization signal (NLS) in order to promote nuclear entry after persistent heat stimulation (90 min at 28°C) (Ippolito et al., 2021). Here, we identified additional functional NES and NLS on CMK-1, whose activity can counteract previously identified elements. Furthermore, we clarify the relationship between the CaM-binding-dependent and T179-dependent effects. T179 phosphorylation can promote nuclear entry both downstream of CaM binding and as part of an independent/parallel pathway. Moreover, T179 phosphorylation can also produce the opposite effect by promoting nuclear export. Taken together, our studies suggest that multiple calcium-dependent regulatory mechanisms converge to bias the activity pattern across a network of NES/NLS elements, in order to control CMK-1 nucleo-cytoplasmic shuttling, and actuate stimulation-dependent nociceptive plasticity.

## Editor's evaluation

In this follow-up study to their previous work (Ippolito, 2021), the authors report additional insights into a complex network of nuclear export (NES) and nuclear localisation (NLS) sequences in the CaM kinase-1 (CMK-1), which is implicated in the plastic regulation of the FLP thermo-nociceptors neurons.

## Introduction

Ca$^{2+}$/CaM-dependent protein kinases (CaMK) are conserved effectors of the universal calcium second messenger, mediating important plasticity effects in the nervous system. In response to neuronal activation, these kinases phosphorylate diverse intracellular substrates in order to actuate changes in cell physiology (*Soderling, 1999*). CaMKI includes an N-terminal kinase domain and a C-terminal regulatory domain, which inhibits its catalytic activity (*Goldberg et al., 1996*; *Hook and Means, 2001*). This auto-inhibition is released upon binding of Ca$^{2+}$/CaM in the regulatory domain and the kinase can be fully activated after phosphorylation of a specific threonine in the activation loop (T177 in human) by CaMK kinase I (CaMKKI). In mammals, CaM binding on CaMKI is a prerequisite prior

*For correspondence:
dominique.glauser@unifr.ch

**Competing interest:** The authors declare that no competing interests exist.

to the phosphorylation of T177. The *Caenorhabditis elegans* CaMKI (named CMK-1) displays similar structural and functional properties, with a conserved activation loop threonine (T179 in the worm sequence) (*Eto et al., 1999*; *Kimura et al., 2002*). CMK-1 mediates multiple plasticity mechanisms, including during salt aversive learning, habituation to repeated mechanical stimulations, and thermal adaptation to innocuous and noxious heat (*Ardiel et al., 2018*; *Lim et al., 2018*; *Neal et al., 2015*; *Satterlee et al., 2004*; *Schild et al., 2014*; *Yu et al., 2014*). The subcellular localization of CMK-1 can be regulated in a stimulus-dependent manner in different contexts (*Moss et al., 2016*; *Neal et al., 2015*; *Schild et al., 2014*; *Yu et al., 2014*).

The role and the mechanisms controlling CMK-1 subcellular localization have been so far best characterized in FLP thermo-nociceptor neurons where CMK-1 regulates nociceptive habituation. Previous experiments conducted in transgenic animals expressing different CMK-1 mutants in FLP with constitutive nuclear or cytoplasmic localization notably showed that CMK-1 activity in the cytoplasm promotes heat avoidance response, while its activity in the nucleus acts antagonistically to reduce heat avoidance response (*Schild et al., 2014*). In wild type worms exposed to innocuous temperature (20°C), CMK-1 resides mostly in the FLP cytoplasm, where it promotes high sensitivity to heat and robust avoidance responses. Upon persistent noxious heat stimulation (90 min at 28°C) CMK-1 progressively translocates in the nucleus, where it acts to reduce the animal heat sensitivity and dampen avoidance responses. In our recent publication (*Ippolito et al., 2021*), we identified two key intrinsic determinants of CMK-1 that control localization in FLP. First, the NES$^{288-294}$ canonical nuclear export sequence of CMK-1, located immediately adjacent to the CaM-binding domain in the regulatory domain and representing a major determinant of CMK-1 cytoplasmic localization at 20°C. Second, the NLS$^{71-78}$ canonical nuclear localization sequence, located on the N-terminal catalytic domain of CMK-1, and which can interact with IMA-3 importin in vitro. We furthermore showed that the binding of Ca$^{2+}$/CaM on CMK-1 can strengthen its interaction with IMA-3 importin via NLS$^{71-78}$ (*Ippolito et al., 2021*), constituting a calcium-dependent mechanism through which thermosensory activity causes CMK-1 nuclear accumulation. Previous studies also suggested a second calcium-dependent mechanism involving the *C. elegans* CaMKKI (named CKK-1) and CMK-1 phosphorylation on T179. Indeed, a T179A mutation on CMK-1 as well as a null mutation in the *ckk-1* gene can both decrease CMK-1 nuclear accumulation at 28°C (*Schild et al., 2014*). However, the mechanisms linking T179 phosphorylation to CMK-1 nuclear entry were still unclear. Furthermore, whereas the NES$^{288-294}$/NLS$^{71-78}$ sequence pair plays a major role, several lines of evidence indicate that additional, uncharacterized elements are also involved. Here, we expand the characterization of the mechanisms controlling CMK-1 localization in FLP neurons to (i) identify remaining CMK-1 elements working as secondary NES and NLS and (ii) better understand the role played by T179 phosphorylation. Our results elucidate how the interplay between multiple, antagonistic pathways are integrated to control CMK-1 localization, and furthermore suggest new regulation opportunities that might be used in other neurons and contexts, in order for CMK-1 localization to integrate different signaling inputs.

## Results and discussion
### Additional elements antagonize NLS$^{71-78}$/NES$^{288-294}$ -dependent regulation of CMK-1 localization

Our previous studies identified NES$^{288-294}$ as the major determinant of CMK-1 cytoplasmic localization at 20°C, and NLS$^{71-78}$ as the main drive for CMK-1 nuclear entry after calcium increase at 28°C (*Figure 1A* schematic) (*Ippolito et al., 2021*). A series of previous observations also suggested that a layer of secondary localization determinants might also play a role (*Ippolito et al., 2021*). The presence of additional functional elements is most strikingly illustrated in an NLS$^{71-78}$; NES$^{288-294}$ double mutant lacking the two main regulatory regions. This double mutant is slightly enriched in the nucleus at 20°C and translocates in the cytoplasm at 28°C, corresponding to a reverted behavior as compared to wild type CMK-1 (*Figure 1B*). The simplest explanation for this behavior would involve a secondary pair of antagonistic NES/NLS determinants, whose activity becomes predominant only when the major drives (NES$^{288-294}$/NLS$^{71-78}$ -dependent pathways) are deactivated. In order to identify these secondary elements, we conducted further mutagenesis analyses in the CMK-1 NLS$^{71-78}$; NES$^{288-294}$ double mutant background.

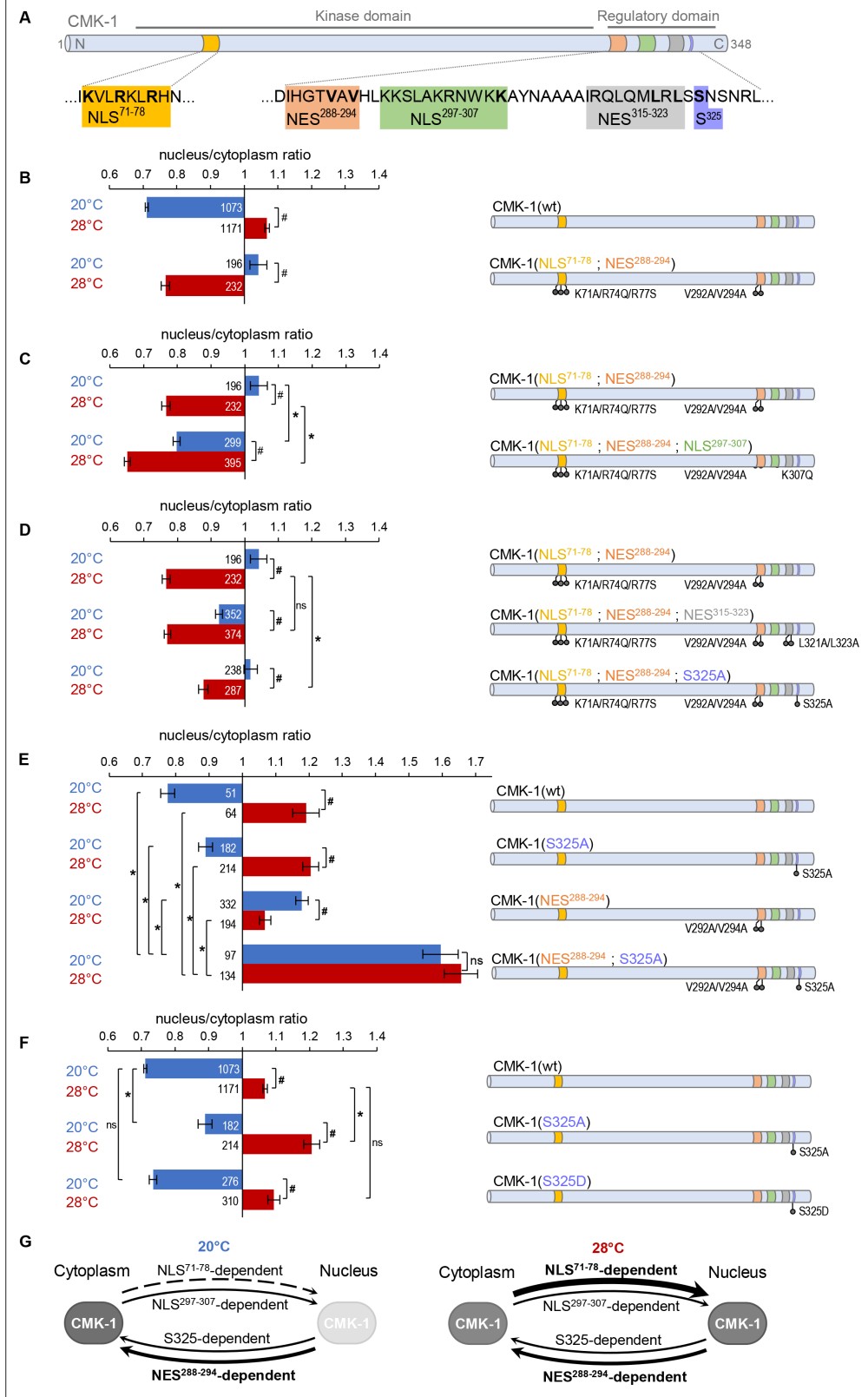

**Figure 1.** Multiple nuclear export sequence (NES) and nuclear localization signal (NLS) elements work in concert to regulate CaM kinase-1 (CMK-1) localization. (**A**) Schematic of CMK-1 with positions of the tested NES and NLS candidates and their sequence. (**B–F**) Subcellular localization of CMK-1::mNeonGreen reporters expressed in FLP and scored after 90 min at 20°C (blue) or at 28°C (red). Average nuclear/cytoplasm fluorescent signal ratio (

*Figure 1 continued on next page*

*Figure 1 continued*

± SEM, left) of wild type (wt) CMK-1 or indicated mutants as schematized (right). Detailed data distributions and ANOVA results are presented in **Figure 1—figure supplement 1**. #p<0.01 between temperature conditions; *p<0.01 between indicated genotypes; ns, not significant by Bonferroni post hoc tests. The numbers of animals scored in each condition (**n**) are indicated for each bar. Datasets for CMK-1(wt) and CMK-1(NLS[71-78]; NES[288-294]) are common across panels. (**G**) Updated model of the multiple elements controlling CMK-1 nuclear export and import. At 20°C, NLS[71-78]-dependent entry is not active (dashed arrow), leaving NES[288-294] as the main drive for cytoplasmic accumulation (thick arrow). At 28°C, NLS[71-78]-dependent entry is active (thickest arrow), becoming the predominant drive to shift CMK-1 equilibrium toward the nucleus. The NLS[297-307]-dependent nuclear entry pathway and the S325-dependent cytoplasmic accumulation favoring pathway are two secondary pathways (thin arrows), whose activity mostly manifests when one or more of the remaining elements are impaired. For simplicity, the S325-dependent pathway is schematized like an export pathway. However, the S325-dependent pathway could also work via an enhanced cytoplasmic retention.

The online version of this article includes the following source data and figure supplement(s) for figure 1:

**Source data 1.** Nuclear/cytoplasmic CaM kinase-1 (CMK-1) expression ratio raw data for **Figure 1**.

**Figure supplement 1.** CaM kinase-1 (CMK-1) localization data distributions and ANOVA results.

Our first goal was to identify the NLS element that could be active at 20°C and promote the nuclear accumulation of the NLS[71-78]; NES[288-294] double mutant. We previously identified NLS[297-307] as a secondary canonical NLS making a minor contribution to the heat stimulus-evoked nuclear accumulation of CMK-1 (**Ippolito et al., 2021**). We thus wondered if NLS[297-307] could be responsible for the nuclear accumulation of the NLS[71-78]; NES[288-294] double mutant in FLP at 20°C. We found that impairing the NLS[297-307] element with a K307Q mutation in the NLS[71-78]; NES[288-294] double mutant background fully blocked its nuclear accumulation at 20°C (**Figure 1C**). This mutation could also reinforce the cytoplasmic enrichment at 28°C (**Figure 1C**). Therefore, we conclude that the NLS[297-307]-dependent nuclear entry pathway contributes to translocate CMK-1 in the nucleus, independently of the NLS[71-78]/NES[288-294] pair.

Our next goal was to identify the secondary element responsible for the heat stimulus-evoked cytoplasmic translocation of the NLS[71-78]; NES[288-294] double mutant. Our first candidate was the putative NES[315-323]. This candidate element was dispensable for cytoplasmic accumulation of wild type CMK-1, in which NES[288-294] is intact (**Ippolito et al., 2021**), but could potentially take over CMK-1 export when NES[288-294] is lacking. We found that the NLS[71-78]; NES[288-294]; NES[315-323] triple mutant protein could still be exported from the nucleus, like the NLS[71-78]; NES[288-294] double (**Figure 1D**). Therefore, the candidate NES[315-323] sequence does not mediate CMK-1 nuclear export in any of the conditions examined so far across our studies. In silico analysis tools focusing on the canonical exportin recognition motif (**Fu et al., 2011**; **Xu et al., 2015**) did not predict additional candidate NESs. To delineate additional candidates, we focused on the C-terminal region of CMK-1. Indeed, a CMK-1 (1–304) mutant was previously shown to accumulate in the nucleus and proposed to lack some NES located in the 305–348 region (**Schild et al., 2014**). Since we can now exclude the NES[315-323] sequence as a functional canonical NES, we looked for additional candidate residues as part of a non-canonical cytoplasmic localization-promoting element in this region. The subcellular localization of protein is often regulated by their phosphorylation and there are precedents for this type of regulation among CaM kinases (**Shioda and Fukunaga, 2018**). Among five S/T residues in the C-terminal region of CMK-1, we focused on S325 which is a strong predicted phosphorylation site as part of an unstructured region (NetPhos3.1, **Blom et al., 2004**) and which matches the prevalent RXXS target substrate motif shared by multiple kinases (**Bradley and Beltrao, 2019**). We found that an S325A mutation produced a partial yet significant reduction in heat stimulus-evoked nuclear export when introduced in the NLS[71-78]; NES[288-294] double mutant background (**Figure 1D**). Furthermore, as compared to CMK-1(wt), the single S325A mutant displayed a significantly more nuclear localization at both 20°C and 28°C (**Figure 1E**). A double mutant protein combining S325A and NES[288-294] mutations markedly accumulated in the nucleus at both temperatures, reaching nuclear/cytoplasmic signal ratio values significantly higher than that of any single mutation (**Figure 1E**). Therefore, serine 325 is part of a functional element promoting CMK-1 cytoplasmic localization and regulating CMK-1 localization independently of NES[288-294].

The S325A mutation that we examined so far is a 'phosphorylation-dead' mutation. If S325 can indeed be phosphorylated and if this modification regulates CMK-1 localization, then an S325D phosphomimic mutation should not have the same impact as the S325A mutation. Consistent with this view, the S325D mutant localization was the same as wild type (*Figure 1F*). These observations suggest (i) that an intact S325 residue is necessary for the normal nuclear export of CMK-1(wt) and (ii) that S325 phosphorylation might be the default state of CMK-1 in FLP under our conditions. Our data also suggest that the post-translational modification of S325 could be used as a regulatory switch to control CMK-1 localization. Nevertheless, we have so far no evidence that this latter mechanism is engaged in heat stimulus-evoked CMK-1 localization in FLP. We know that S325 promotes CMK-1 cytoplasmic accumulation independently of NES[288-294] (as shown by the impact of the S325A mutation in a NES[288-294] mutant background *Figure 1D and E*), but how the S325-element works remains to be determined. In principle, it could either function by promoting a non-canonical nuclear export or a cytoplasmic retention pathway, for example, by anchoring CMK-1 to some cytoplasmic components. Furthermore, it seems that S325 phosphorylation/dephosphorylation modifications are not involved in the heat-evoked CMK-1 re-localization process in FLP. However, we speculate that this mechanism might be relevant in other contexts in FLP and/or in other neuron types.

## An updated model of the functional NES and NLS elements of CMK-1

Together with the results of our previous report (*Ippolito et al., 2021*), our data show that CMK-1 subcellular localization is regulated by multiple intrinsic elements:

i.   NES[288-294], working via the canonical exportin pathway as primary nuclear export drive at 20°C
ii.  NLS[71-78], working via a canonical IMA-3-dependent pathway as primary cell stimulus-dependent nuclear import drive at 28°C
iii. NLS[297-307], as a secondary pathway working in partial redundancy with NLS[71-78], most likely via a canonical importin pathway
iv.  the S325 phosphosite in the C-terminal portion of CMK-1, which is part of a cytoplasmic localization-promoting pathway presumably active when S325 is phosphorylated.

The schematic in *Figure 1G* presents a working model of the regulation of wild type CMK-1 localization with the relative contribution of each pathway at 20°C and 28°C. At 20°C, the strong NLS[71-78]-dependent nuclear entry pathway is inactive and the activity of the weaker NLS[297-307] pathway is over-competed by both NES[288-294]-dependent and the S325-dependent export pathways, which are both active. At 28°C, the strong NLS[71-78]-dependent nuclear entry pathway is active and is the main drive to shift the equilibrium toward the nucleus, even if the other elements might still be active.

## CMK-1 T179 phosphorylation is required for CMK-1 nuclear entry downstream of intracellular calcium elevation

Previous studies have shown that a phosphorylation-dead CMK-1(T179A) mutant protein displays impaired activity-dependent nuclear accumulation (*Schild et al., 2014*; *Yu et al., 2014*). We recapitulated these previous findings using a CMK-1::mNeonGreen reporter system (*Hostettler et al., 2017*). Indeed, while the T179A mutation left CMK-1 reporter localization unchanged in FLP at 20°C, it abolished its nuclear accumulation after 90 min of cell activation at 28°C (*Figure 2A–B*). Hence, we confirm that the ability of CMK-1 to be phosphorylated on T179 is essential for its activity-dependent nuclear translocation.

These results are in line with a simple model in which a prolonged temperature elevation, mirrored by prolonged intracellular calcium elevation in FLP (*Saro et al., 2020*), will activate CKK-1, which will phosphorylate CMK-1 as a necessary step for nuclear accumulation. To confirm that T179-dependent nuclear accumulation indeed takes place downstream of calcium elevation, we tested if the T179A mutation could block the effect of the *unc-68(dom13)* mutation, which chronically elevates FLP cytoplasmic calcium levels at both 20°C and 28°C to promote CMK-1 nuclear accumulation (*Figure 2A*; *Ippolito et al., 2021*; *Marques et al., 2019*). The CMK-1 nuclear accumulation caused by the *unc-68(dom13)* mutation was fully abolished by the T179A mutation at either temperature (*Figure 2C*). We conclude that the phosphorylation of T179 mediates the nuclear accumulation of CMK-1 downstream of $Ca^{2+}$ elevation, irrespective of the temperature.

Next, we assessed the interaction between T179A and a W305S mutation preventing CaM binding on CMK-1. Each of these mutations alone or in combination had little impact on the cytoplasmic

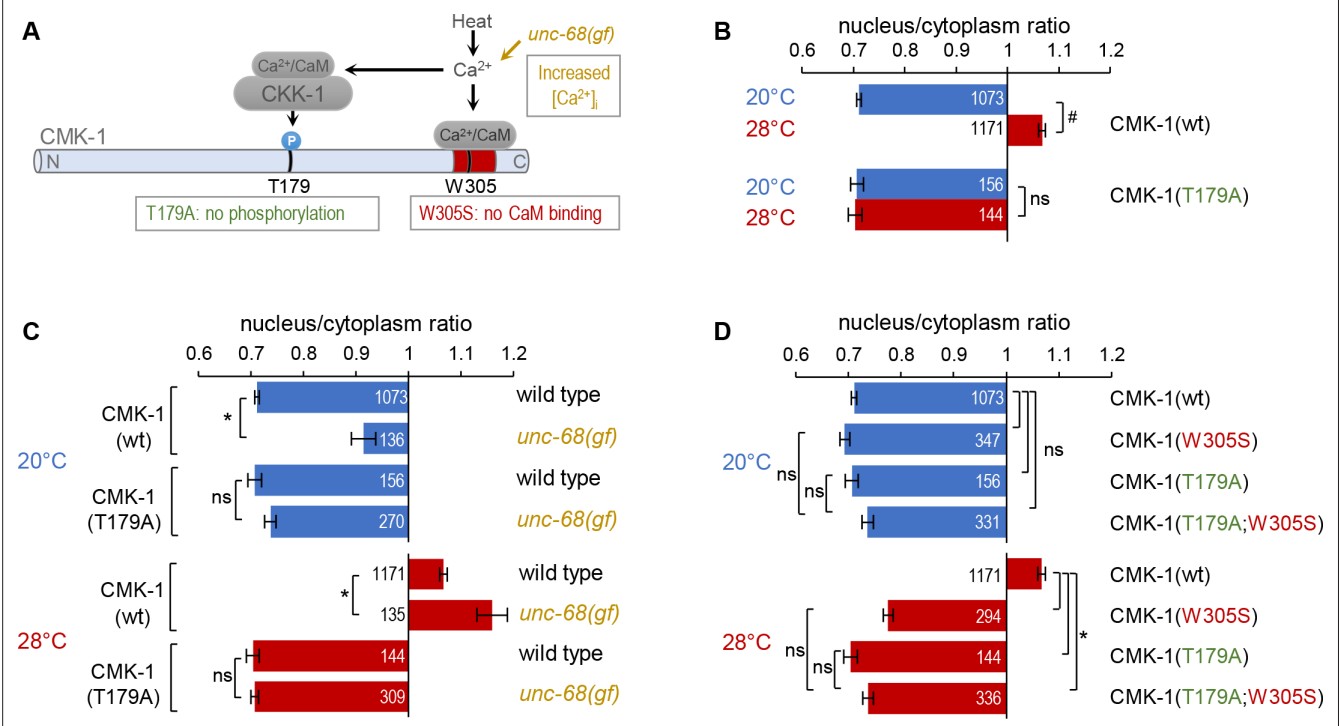

**Figure 2.** CaM kinase-1 (CMK-1) T179 phosphorylation is required for CMK-1 nuclear entry downstream of intracellular calcium elevation. (**A**) Schematic of calcium-dependent CMK-1 activation pathway involving CaM binding and CaM kinase kinase-1 (CKK-1) phosphorylation on T179. Illustration of the tested mutations and their effects. (**B–D**) Subcellular localization of CMK-1::mNeonGreen reporters expressed in FLP and scored after 90 min at 20°C (blue) or at 28°C (red). Average nuclear/cytoplasm fluorescent signal ratio ( ± SEM) of wild type (wt) CMK-1 or indicated mutants. Detailed data distributions and ANOVA results are presented in *Figure 2—figure supplement 1*. #p<0.01 between temperature conditions; *p<0.01 between indicated genotypes; ns, not significant by Bonferroni post hoc tests. The numbers of animals scored in each condition (**n**) are indicated for each bar. Datasets for CMK-1(wt) and CMK-1(T179A) are common across panels and CMK-1(wt) dataset is the same as in *Figure 1*.

The online version of this article includes the following source data and figure supplement(s) for figure 2:

**Source data 1.** Nuclear/cytoplasmic CaM kinase-1 (CMK-1) expression ratio raw data for *Figure 2*.

**Figure supplement 1.** CaM kinase-1 (CMK-1) localization data distributions and ANOVA results.

localization of CMK-1 at 20°C (*Figure 2D*, top). At 28°C, the single mutations as well as the combined mutations had a similar effect, fully preventing the nuclear accumulation of CMK-1, with ratio values similar to the situation at 20°C (*Figure 2D*). These results confirm previous findings indicating that T179 phosphorylation and CaM binding are two major events controlling heat-evoked CMK-1 nuclear accumulation. Whereas we cannot rule out a floor effect in the ratio measure, the lack of a cumulative effect between the two mutations suggests that CaM binding and T179 phosphorylation might be part of the same pathway.

Collectively, these results are consistent with a model in which T179 phosphorylation acts downstream of intracellular calcium elevation and CaM binding to regulate cell activity-dependent CMK-1 nuclear entry.

## CMK-1 T179 phosphorylation favors nuclear translocation at 20°C independently of CaM binding

Having shown that T179 phosphorylation is required for CMK-1 nuclear accumulation at 28°C, we next wondered if it could be sufficient to trigger this re-localization event in the absence of cell stimulation and of CaM binding. To address this question, we examined the impact of the T179D phospho-mimicking mutation at 20°C and found it could favor nuclear accumulation in the absence of thermal stimulation (*Figure 3A and B*). It is however important to note that the effect of the T179D mutation at 20°C was not as strong as the impact of heat on CMK-1(wt). Therefore, whereas T179 phosphorylation

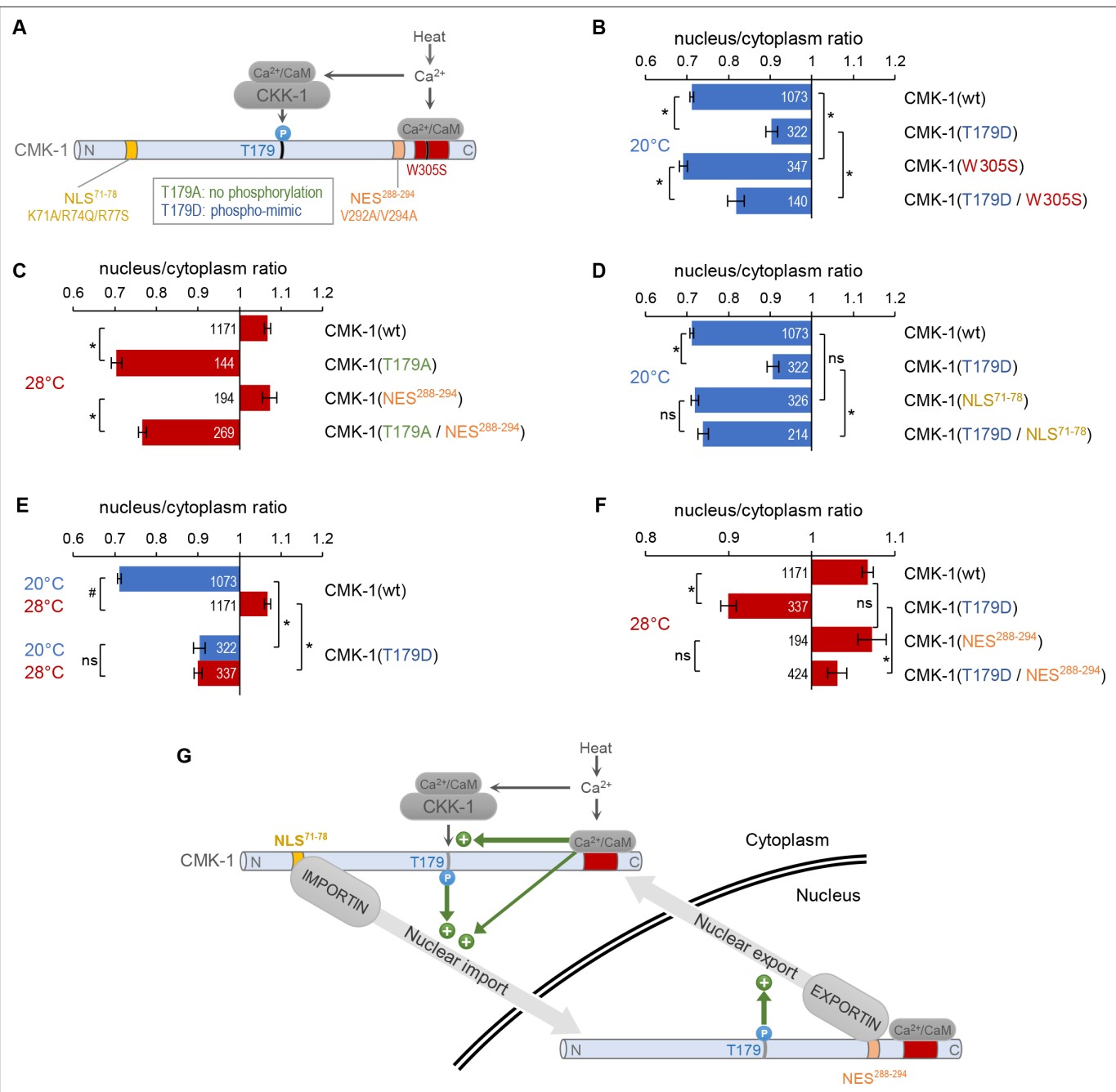

**Figure 3.** CaM kinase-1 (CMK-1) T179 phosphorylation promotes both NLS[71-78]-dependent import and NES[288-294]-dependent export. (**A**) Schematic of calcium-dependent CMK-1 activation pathways involving CaM binding and CaM kinase kinase-1 (CKK-1) phosphorylation on T179. Illustration of the tested mutations and their effects. (**B–F**) Subcellular localization of CMK-1::mNeonGreen reporters expressed in FLP and scored after 90 min at 20°C (blue) or at 28°C (red). Average nuclear/cytoplasm fluorescent signal ratio ( ± SEM) of wild type (wt) CMK-1 or indicated mutants. Detailed data distributions and ANOVA results are presented in *Figure 3—figure supplement 1*. [#]p<0.01 between temperature conditions; *p<0.01 between indicated genotypes; ns, not significant by Bonferroni post hoc tests. The numbers of animals scored in each condition (**n**) are indicated for each bar. Datasets for CMK-1(wt) and CMK-1(T179D) are common across panels and CMK-1(wt) dataset is the same as in *Figures 1 and 2*. (**G**) Schematic of the multiple cell stimulation-dependent pathways occurring downstream of calcium elevation (green arrows) and proposed to control CMK-1 nucleo-cytoplasmic shuttling. CaM binding to CMK-1 can (**i**) directly promote nuclear import via NLS[71-78]-dependent pathway, fostered by an enhanced affinity for IMA-3 and (ii) favor CKK-1-dependent phosphorylation of T179. In turn T179 phosphorylation has a dual effect. First, it promotes NLS[71-78]-dependent nuclear entry. Second, it promotes NES[288-294]-dependent nuclear export. This mechanism is proposed to sustain an enhanced nucleo-cytoplasmic shuttling activity and the progressive equilibrium shift toward the nucleus after prolonged heat-evoked FLP stimulation.

The online version of this article includes the following source data and figure supplement(s) for figure 3:

*Figure 3 continued on next page*

*Figure 3 continued*

**Source data 1.** Nuclear/cytoplasmic CaM kinase-1 (CMK-1) expression ratio raw data for *Figure 3*.

**Figure supplement 1.** CaM kinase-1 (CMK-1) localization data distributions and ANOVA results.

directly contribute to trigger CMK-1(wt) nuclear accumulation at 28°C, additional mechanisms are likely to occur.

At 20°C, the T179D mutation could also favor CMK-1 nuclear localization in a W305S mutant background abolishing CaM binding (*Figure 3B*; two-way ANOVA: main effect of T179D, p<0.001; main effect of W305S, p<0.001, T179D × W305S interaction, p=0.637). Thus, the T179D phospho-mimic mutation is sufficient to promote CMK-1 nuclear accumulation independently of CaM binding at 20°C, suggesting that T179 phosphorylation functions either downstream of CaM binding or in a separate pathway. Interestingly, our data also show that the W305S mutation produced a small cytoplasmic translocation effect in both wild type and T179D backgrounds (*Figure 3B*). These observations are compatible with the results of our previous protein interaction analyses, having shown that CaM binding can increase CMK-1 affinity for IMA-3 in an in vitro context devoid of CKK-1 and of T179 phosphorylation (*Ippolito et al., 2021*). The simplest model explaining these results would involve two mechanisms through which CaM binding promotes CMK-1 nuclear accumulation: (i) a direct pathway, through which CaM binding reinforces the interaction between IMA-3 and the NLS[71-78] element of CMK-1 and (ii) an indirect pathway, through which CaM binding is required prior to phosphorylation of T179, which in turn promotes the nuclear accumulation of CMK-1.

## CMK-1 T179 phosphorylation causes NLS[71-78]-dependent nuclear translocation at 20°C

Our next goal was to determine how T179 phosphorylation could promote CMK-1 nuclear accumulation. We tested two hypothetical models implicating either NLS[71-78] or NES[288-294] because these two elements are the main regulators of stimulation-dependent CMK-1 localization. In the first model, T179 phosphorylation would act by masking NES[288-294]. In the second model, T179 phosphorylation would act by unmasking NLS[71-78].

To test the first model, we examined the interaction between T179A and NES[288-294] mutations. We found that T179A markedly promoted CMK-1 cytoplasmic retention at 28°C regardless of the NES[288-294] mutation (*Figure 3C*, two-way ANOVA: main effect of T179A, p<0.001; main effect of NES[288-294], p<0.001, T179A × NES[288-294] interaction, p=0.384). Therefore, it is unlikely that the T179A mutation predominantly causes CMK-1 cytoplasmic translocation by aberrantly revealing NES[288-294] at 28°C.

To test the second model, we examined the interaction between T179D and NLS[71-78] mutations. The T179D nuclear translocation effect at 20°C was abolished in a T179D; NLS[71-78] double mutant (*Figure 3D*, significant interaction effect). Therefore, T179D phosphomimic mutation promotes NLS[71-78]-dependent CMK-1 nuclear entry.

Taken together with our previous finding (*Ippolito et al., 2021*), our new results suggest a model in which multiple calcium-dependent events in the cytoplasm converge on CMK-1 NLS[71-78] element to activate its nuclear import upon prolonged cell activation (see cytoplasmic green arrows in *Figure 3G*).

## CMK-1 T179 phosphorylation favors cytoplasmic localization in a NES[288-294]-dependent manner at 28°C

As mentioned above, the nuclear accumulation caused by the T179D mutation at 20°C is not as marked as the one caused on wild type CMK-1 by the cell activation at 28°C. We considered two hypothetical explanations for this difference. On the one hand, the lack of CaM binding at 20°C might prevent a fully efficient nuclear entry despite the presence of the T179D mutation. On the other hand, the chronic phosphorylation-like state in the T179D mutant might also trigger an antagonist mechanism concomitantly promoting CMK-1 cytoplasmic translocation, and therefore tempering the nuclear accumulation of the T179D mutant protein. If the first model is true, one would predict the T179D mutant to further accumulate in the nucleus when the cell is activated at 28°C and $Ca^{2+}$/CaM available for CMK-1 binding. In strong contrast to this prediction, CMK-1(T179D) mutant protein at 28°C was not more nuclear than CMK-1(wt), but actually significantly more cytoplasmic with a ratio similar to that of CMK-1(T179D) at 20°C (*Figure 3E*). Therefore, the impact of T179D mutation is

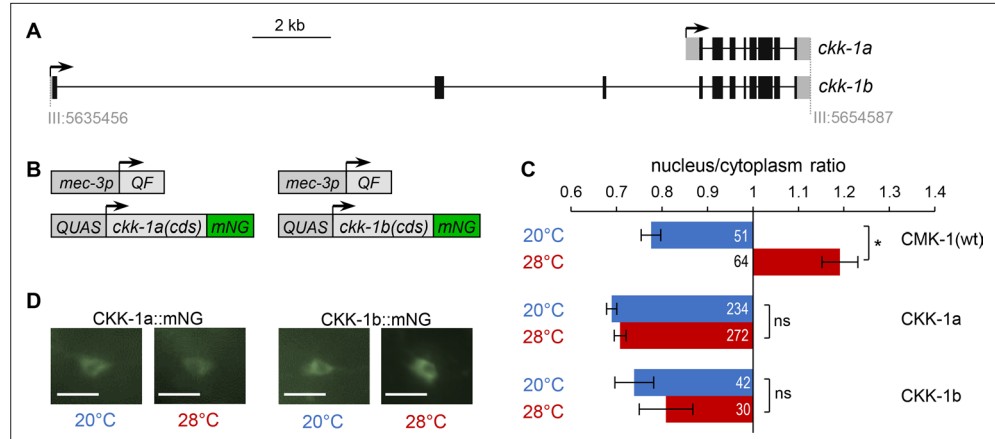

**Figure 4.** CaM kinase kinase-1 (CKK-1) subcellular localization in FLP. (**A**) Schematic of *ckk-1* gene locus showing genomic coordinates and the two predicted isoforms produced via alternative transcription start. Coding exon (black boxes), untranslated regions (gray boxes). (**B**) Schematic of the Q-system-based construct combinations used to drive mNeonGreen (mNG) fusion reporters for CKK-1a (B, left) and CKK-1b (B, right), respectively. (**C**) Subcellular localization of mNeonGreen reporters expressed in FLP and scored after 90 min at 20°C (blue) or at 28°C (red). Average nuclear/cytoplasm fluorescent signal ratio ( ± SEM) for the indicated reporters. Detailed data distributions presented in *Figure 4—figure supplement 1*. [#]$p<0.01$ between temperature conditions by Student's *t*-test; ns, not significant. CMK-1 ratios are presented for comparison purpose. The numbers of animals scored in each condition (**n**) are indicated for each bar. (**D**) Representative fluorescence micrographs showing FLP signals. The nucleus is visible as a low-signal region (dark). Scale bars: 10 µm.

The online version of this article includes the following source data and figure supplement(s) for figure 4:

**Source data 1.** Nuclear/cytoplasmic CaM kinase-1 (CMK-1) and CaM kinase kinase-1 (CKK-1) expression ratio raw data for *Figure 4*.

**Figure supplement 1.** CaM kinase kinase-1 (CKK-1) localization data distributions and ANOVA results.

context-dependent, and, in addition to its nuclear entry promoting effect revealed at 20°C, might also include an antagonistic nuclear export drive, which becomes apparent at 28°C to counteract the nuclear accumulation. In order to determine if T179D favors nuclear export via the NES[288-294] element, we next examined the localization of double mutant proteins combining T179D with NES[288-294]-impairing mutations. We found that the T179D-evoked cytoplasmic drive was strongly reduced in the absence of an intact NES[288-294] element (*Figure 3F*). We conclude that the T179D phospho-mimic mutation promotes CMK-1 nuclear export in a NES[288-294]-dependent manner.

Taken together, our data suggest a model in which prolonged FLP stimulation causes T179 phosphorylation, which activates both NLS[71-78]-dependent nuclear entry (*Figure 3G*, cytoplasmic green arrow) and NES[288-294]-dependent export pathways (*Figure 3G*, nuclear green arrow). T179 phosphorylation would thus result in (i) an activated dynamic nucleo-cytoplasmic shuttling and (ii) a shift in CMK-1 equilibrium toward the nucleus. The latter effect most likely results from a synergy between the direct impact of T179 phosphorylation and the Ca[2+]/CaM-binding-evoked importin affinity increase, converging on NLS[71-78] in order for this pathway to dominate over the antagonistic pathways into play.

## Both CKK-1 isoforms localize to the cytoplasm of FLP regardless of heat stimulation

Our next goal was to identify in which cellular compartment CMK-1 is most likely to be phosphorylated by CKK-1. In principle, T179 phosphorylation may take place in the cytoplasm, in the nucleus or in both compartments, depending on the localization of CKK-1. In rats, CaM kinase kinase alpha is expressed in the nucleus, whereas CaM kinase kinase beta is expressed in the nucleus and/or the cytoplasm depending on the tissue (*Kitani et al., 2003*). The worm genome encodes two predicted CKK-1 isoforms (CKK-1a and CKK-1b), produced via alternative transcription start (*Figure 4A*), which diverge in their N-terminal region. Their subcellular localization was unknown. Using different in silico prediction tools (*Fu et al., 2011*; *Nguyen Ba et al., 2009*; *Xu et al., 2015*), we did not identify any

consensus NLS in either isoform, but only a weak predicted NES element, which suggested that CKK-1 might be excluded from the nucleus. To test this prediction empirically, we generated mNG protein fusion reporter for CKK-1a and -1b isoforms, respectively, and targeted their expression in FLP with the Q-system (*Figure 4B*; *Schild and Glauser, 2015*; *Wei et al., 2012*). Both isoforms were markedly enriched in the cytoplasm, with ratios similar to that of CMK-1 at rest (*Figure 4C and D*). Furthermore, their localization was not altered upon heat stimulation at 28°C for 90 min. These data suggest that most of CKK-1 activity in FLP is likely to occur in the cytoplasm and that CMK-1 phosphorylation might therefore primarily take place in this compartment (as depicted in *Figure 3G*).

## Perspective and conclusion

CMK-1 subcellular localization contributes to adjust sensory neuron activity according to past experience. Heat-evoked CMK-1 nuclear entry in FLP thermo-nociceptor neurons causes a decrease in the animal responsiveness to noxious thermal stimuli (*Schild et al., 2014*). As expected for an intracellular signaling event able to modify animal behavior and survival prospects, CMK-1 subcellular localization control is a tightly regulated process, which needs to take place only in a specific context and over an appropriate timescale. Our studies have identified multiple functional intrinsic elements, including NES, NLS, and phosphorylation sites, and provide a more complete picture of their interplay in FLP thermo-nociceptor neurons. These elements have antagonistic effects, favoring either export or import, and in the case of the T179 phosphorylation, favoring both export and import. We speculate that the dual regulatory role of T179 phosphorylation in promoting bidirectional CMK-1 translocation across the nuclear envelope could be at the origin of cell stimulation-dependent CMK-1 shuttling cycles. Importin/exportin-based transport is an energy-consuming process. Therefore, if a single phosphorylation event promotes the two antagonistic pathways, it must also come with some benefits. Considering the cytoplasmic localization of CKK-1, we propose that a dynamic nucleo-cytoplasmic shuttling might contribute to ensure that the T179 phosphorylation status of CMK-1 is frequently 'refreshed', in order for the nuclear pool of CMK-1 to reflect the current activation status of CKK-1 in the cytoplasm. To test this hypothetical model, future studies will need to quantify the actual shuttling kinetics of wild type and mutant CMK-1 at different temperatures. Furthermore, the whole process is likely to be also influenced by the activity of phosphatase(s) responsible for CMK-1 dephosphorylation on T179. In order to obtain a full picture of the phosphorylation-dependent CMK-1 localization regulation, future studies will also need to identify these phosphatases and determine their subcellular locus of action in vivo.

Previous studies have reported transient, persistent, or sometimes no CMK-1 nuclear translocation in different neurons subjected to different stimulation regimens (*Ippolito et al., 2021*; *Lim et al., 2018*; *Moss et al., 2016*; *Schild et al., 2014*; *Yu et al., 2014*). We propose that the network of intrinsic regulatory elements that we have identified might be the target of multiple intracellular signaling events in order to fine-tune CMK-1 localization in a variety of contexts. While CaM binding and T179 phosphorylation converging on NLS[71-78] and the antagonistic NES[288-294]-dependent export constitutes the primary mechanism explaining the CMK-1 equilibrium shift upon prolonged stimulation in FLP (*Figure 3G*), the other elements could be more relevant in different contexts. The fact that NLS[297-307] overlaps with the CaM-binding site and the fact that S325 phosphorylation status affects its NES activity represent additional regulation opportunities, which could be engaged in different cells or situations. We speculate that, for example, this could be the case in cells with limited CKK-1 activity, like cells with little or no CKK-1 expression. Indeed, based on previous reporter analyses and single-cell RNA seq data, the expression of CMK-1 is much broader than that of CKK-1 (*Kimura et al., 2002*; *Taylor et al., 2021*). The activity of CaM kinase kinase in mammals was also shown to be regulated by post-translational modifications (*Green et al., 2011*; *Nakanishi et al., 2017*; *Takabatake et al., 2019*) and this pathway may be silenced in specific situations.

Being downstream effectors of intracellular calcium signaling, CaM kinases can couple cell activation with physiological changes taking place over relatively long timescales. Furthermore, multiple studies have shown that their kinase activity can be regulated by other signaling pathways (see *Swulius and Waxham, 2008*; *Tokumitsu and Sakagami, 2022* for reviews), placing them as a hub for integrating multiple inputs. Our study supports the notion that not only CaM kinase activity, but also the regulation of its subcellular localization are important targets of convergent regulatory inputs. Owing to the large conservation of calcium signaling during evolution and in particular of the CaM kinase

structural and functional architectures, similar mechanisms might be relevant for their regulation in different species and for different sensory modalities beyond thermo-nociceptive functions.

# Materials and methods

## Key resources table

| Reagent type (species) or resource | Designation | Source or reference | Identifiers | Additional information |
|---|---|---|---|---|
| Genetic reagent (*C. elegans*) | DAG439 | *Ippolito et al., 2021* | domSi439[mec-3p::cmk-1 (1–348)::mNG::unc-54 3'UTR] II | Expression of CMK-1(wt)::mNG in FLP |
| Genetic reagent (*C. elegans*) | DAG703-704-705 | *Ippolito et al., 2021* | domEx703-704-705[mec-3p::cmk-1(V292A/V294A)::mNG, unc-122p::RFP] | Expression of CMK-1(V292A/V294A)::mNG in FLP |
| Genetic reagent (*C. elegans*) | DAG1032 | *Ippolito et al., 2021* | domSi439[mec-3p::cmk-1::mNG::3xFlag::unc-54 3'UTR] II;[unc-68(dom13)] V | Expression of CMK-1(wt)::mNG in FLP in *unc-68* mutant background |
| Genetic reagent (*C. elegans*) | DAG900-901-902 | *Ippolito et al., 2021* | domEx900-901-902[mec-3p::cmk-1(K71A/R74Q/R77S/V292A/V294A)::mNG, unc-122p::RFP] | Expression of CMK-1(K71A/R74Q/R77S/V292A/V294A)::mNG in FLP |
| Genetic reagent (*C. elegans*) | DAG700-701-702 | *Ippolito et al., 2021* | domEx700-701-702[mec-3p::cmk-1(W305S)::mNG, unc-122p::RFP] | Expression of CMK-1(W305S)::mNG in FLP |
| Genetic reagent (*C. elegans*) | DAG706-707-708 | *Ippolito et al., 2021* | domEx706-707-708[mec-3p::cmk-1(K71A/R74Q/R77S)::mNG, unc-122p::RFP] | Expression of CMK-1(K71A/R74Q/R77S)::mNG in FLP |
| Genetic reagent (*C. elegans*) | DAG709-710-711-1134-1135-1136 | This study | domEx709-710-711-1134-1135-1136 [mec-3p::cmk-1(T179D)::mNeonGreen; unc-122p::RFP] | Expression of CMK-1(T179D)::mNG in FLP |
| Genetic reagent (*C. elegans*) | DAG744-745-746-812-813 | This study | domEx744-745-746-812-813[mec-3p::cmk-1(T179A)::mNeonGreen; unc-122p::RFP] | Expression of CMK-1(T179A)::mNG in FLP |
| Genetic reagent (*C. elegans*) | DAG906-907-908 | This study | domEx906-907-908[mec-3p::cmk-1(K71A/R74Q/R77S/ T179D)::mNeonGreen, unc-122p::RFP] | Expression of CMK-1(K71A/R74Q/ R77S/T179D)::mNG in FLP |
| Genetic reagent (*C. elegans*) | DAG909-910-911 | This study | domEx909-910-911[mec-3p::cmk-1(T179D/W305S)::mNeonGreen, unc-122p::RFP] | Expression of CMK-1(W305S/T179D)::mNG in FLP |
| Genetic reagent (*C. elegans*) | DAG924-925-926 | This study | domEx924-925-926[mec-3p::cmk-1(T179A/V292A/V294A)::mNeonGreen, unc-122p::RFP] | Expression of CMK-1(T179A/ V292A/V294A)::mNG in FLP |
| Genetic reagent (*C. elegans*) | DAG1015-1016-1017 | This study | domEx1015-1016-1017[mec-3p::cmk-1(T179A/W305S)::mNeonGreen, unc-122p::RFP] | Expression of CMK-1(W305S/T179A)::mNG in FLP |
| Genetic reagent (*C. elegans*) | DAG1600-1601-1602 | This study | domEx1600-1601-1602[mec-3p::cmk-1(S325D)::mNeonGreen; unc-122p::RFP] | Expression of CMK-1(S325D)::mNG in FLP |
| Genetic reagent (*C. elegans*) | DAG1621-1622 | This study | domEx1621-1622[mec-3p::cmk-1(S325A)::mNeonGreen; unc-122p::RFP] | Expression of CMK-1(S325A)::mNG in FLP |
| Genetic reagent (*C. elegans*) | DAG1727-1728-1729 | This study | domEx1727-1728-1729[mec-3p]::cmk-1(T179D/V292A/V294A)::mNeonGreen; unc-122p::RFP | Expression of CMK-1(T179D/V292A/V294A)::mNG in FLP |
| Genetic reagent (*C. elegans*) | DAG1733-1734-1735 | This study | domEx1733-1734-1735[mec-3p::cmk-1(K71A/R74Q/R77S/V292A/V294A/L321A/L323A)::mNeonGreen; unc-122p::RFP] | Expression of CMK-1(K71A/R74Q/R77S/V292A/V294A/L321A/L323A)::mNG in FLP |
| Genetic reagent (*C. elegans*) | DAG1739-1740-1741 | This study | domEx1739-1740-1741[mec-3p::cmk-1(K71A/R74Q/R77S/V292A/V294A/K307Q)::mNeonGreen; unc-122p::RFP] | Expression of CMK-1(K71A/R74Q/R77S/V292A/V294A/K307Q)::mNG in FLP |
| Genetic reagent (*C. elegans*) | DAG1748-1749-1750 | This study | domEx1748-1749-1750[mec-3p::QF]; [QUAS::ckk-1b::gfp]; [unc-122p::RFP] | Expression of CKK-1b::mNG in FLP |
| Genetic reagent (*C. elegans*) | DAG1751-1752-1753 | This study | domEx1751-1752-1753[mec-3p::QF]; [QUAS::ckk-1a::gfp]; [unc-122p::RFP] | Expression of CKK-1a::mNG in FLP |

*Continued on next page*

*Continued*

| Reagent type (species) or resource | Designation | Source or reference | Identifiers | Additional information |
|---|---|---|---|---|
| Genetic reagent (*C. elegans*) | DAG1793-1794-1795 | This study | *domEx1793-1794-1795[mec-3p::cmk-1(V292A/V294A/S325A)::mNeonGreen; unc-122p::RFP]* | Expression of CMK-1(V292A/V294A/S325A)::mNG in FLP |
| Genetic reagent (*C. elegans*) | DAG1787-1788-1789 | This study | *domEx1787-1788-1789[mec-3p::cmk-1(K71A;R74Q;R77S;V292A;V294A;S325A)::mNG]; [unc-122p::RFP]* | Expression of CMK-1(K71A;R74Q/R77S/V292A/V294A/S325A)::mNG in FLP |
| Genetic reagent (*C. elegans*) | DAG1430-1431-1432 | This study | *domEx1430-1431-1432[mec-3p::cmk-1(T179A)::mNeonGreen, unc-122p::RFP];[unc-68(dom13)]V* | Expression of CMK-1(T179A)::mNG in FLP in unc-68 mutant background |

## *C. elegans* strains and growth conditions

*C. elegans* strains used in this study are reported in the Key resources table. All strains were grown as previously described (*Stiernagle, 2006*) on nematode growth media (NGM) plates with OP50 *Escherichia coli*, at 20°C.

## In silico sequence analysis

NLS predictions were performed with NLStradamus (*Nguyen Ba et al., 2009*), NES predictions with LocNES (*Xu et al., 2015*), as well as NESsential (*Fu et al., 2011*), and phosphosite predictions with NetPhos3.1 (*Blom et al., 2004*).

## Transgenesis

Plasmid DNA was purified with the GenElute HP Plasmid miniprep kit (Sigma) and microinjected in the worm gonad according to a standard protocol (*Evans, 2006*). We used *unc-122p::RFP* as co-injection markers to identify transgenic animals.

## Promoter plasmids (Multi-site Gateway slot 1)

Entry plasmids containing specific promoters were constructed by PCR from N2 genomic DNA, with primers flanked with attB4 and attB1r recombination sites and cloned into pDONR-P4-P1R vector (Invitrogen) by BP recombination. Plasmids and primer sequences are reported in the *Supplementary file 1*.

## Coding sequence plasmids (Multi-site Gateway slot 2)

Entry plasmids containing specific coding DNA sequences were constructed by PCR from N2 cDNA with primers flanked with attB1 and attB2 recombination sites and cloned into pDONR_221 vector (Invitrogen) by BP recombination. Plasmids and primer sequences are listed in *Supplementary file 1*.

## Site-directed mutagenesis

All the point mutations were generated by inverse PCR-based site-directed mutagenesis (*Hemsley et al., 1989*). In brief, whole plasmids (entry plasmids containing *cmk-1* coding DNA sequences) were amplified with the KOD Hot Start DNA Polymerase (Novagen, Merck). Primers were phosphorylated in 5' and were designed to contain the desired point mutation(s) and to hybridize in a divergent and back-to-back manner on the plasmid. After electrophoresis, linear PCR products were purified from agarose gel (1%) with a Zymoclean-Gel DNA Recovery kit (Zymo Research) and circularized with DNA Ligation Kit 'Mighty Mix' (Takara). The plasmid templates, primer sequences, and names of resulting mutation-carrying plasmids are reported in *Supplementary file 1*.

## Imaging of fluorescent CMK-1 reporter protein

### Worm preparation for CMK-1 imaging in FLP

Worms were synchronized according to standard procedure with hypochlorite treatment and grown at 20°C. First-day adult animals were collected from NGM bacterial plates with distilled water, transferred to 1.5 ml microcentrifuge tubes, and washed once with distilled water. Twenty µl of a dense worm suspension were transferred to PCR tubes and incubated in a thermocycler at 20°C or 28°C

for 1.5 hr. Prior to imaging, worms were immobilized with the addition of $NaN_3$ (final concentration 1% m/v), transferred on a glass slide, and covered with a coverslip. Imaging was carried out during the next 5 min.

## Microscopy
FLP images to measure the nuclear/cytoplasmic ratio were acquired in a Zeiss Axioplan2 fluorescence microscope, with a 40× (air, NA = 0.95) objective and constant illumination parameters.

## Determination of CMK-1 nuclear/cytoplasmic ratio
For CMK-1 subcellular localization analysis, the intensity of fluorescence was first measured for each neuron in three regions of interest (ROIs): nucleus, cytoplasm, background. The nuclear/cytoplasmic ratio was calculated as (nucleus-background)/(cytoplasm-background). A ratio >1 indicates a nuclear accumulation of CMK-1, while a value <1 a cytoplasmic biased ratio. All three ROIs were ellipses of the same area. The background ROI was defined in a worm region close to the neuron to take autofluorescence into account. Nuclear and cytoplasmic ROIs were defined based on the mNG green signal and the shape of the neuron, via a previously validated procedure (*Ippolito et al., 2021*).

## Statistical tests
Comparisons were made with one-way and two-way ANOVAs followed by Bonferroni post hoc tests using Jamovi (The jamovi project (2021). jamovi (Version 1.6) [Computer Software]. Retrieved from https://www.jamovi.org). A visual inspection of Q-Q plots suggested that all datasets were following a normal or nearly normal distribution, but some datasets returned significant results with the Shapiro-Wilk test ($p < 0.01$). For that reason, we conducted robust ANOVAs. No outlier was excluded.

## Experimental design
No specific a priori power analysis was performed. N determination was based on previous similar experiments and adjustments made based on the actual measure variability. All Ns are mentioned in the figures. Ns represent independent animals (biological replicates). At least two independent transgenic lines (in most cases three lines or more, see Key resources table) were scored for each genotype, each on at least three different experimental days. Wild type control was systematically run in parallel. Thermal stimulation conditions at 20°C and 28°C were run in parallel during imaging session. Some wild type control data are reused across some figure panels for datasets matching the same acquisition sessions.

## Materials availability statement
Strains and plasmids generated during the study will be made available upon request.

# Acknowledgements
We are grateful to Lisa Schild and Laurence Bulliard for expert technical support, and to Marc Hammarlund, Bill Schafer, Piali Sengupta, and Miriam Goodman for the gift of plasmids and strains. Some strains were provided by the CGC, which is funded by NIH Office of Research Infrastructure Programs (P40 OD010440). The study was supported by the Swiss National Science Foundation (BSSGI0_155764, PP00P3_150681, and 310030_197607 to DAG).

# Additional information

### Funding

| Funder | Grant reference number | Author |
| --- | --- | --- |
| Schweizerischer Nationalfonds zur Förderung der Wissenschaftlichen Forschung | BSSGI0_155764 | Dominique A Glauser |

| Funder | Grant reference number | Author |
|---|---|---|
| Schweizerischer Nationalfonds zur Förderung der Wissenschaftlichen Forschung | PP00P3_150681 | Dominique A Glauser |
| Schweizerischer Nationalfonds zur Förderung der Wissenschaftlichen Forschung | 310030_197607 | Dominique A Glauser |

The funders had no role in study design, data collection and interpretation, or the decision to submit the work for publication.

## Author contributions

Domenica Ippolito, Conceptualization, Data curation, Formal analysis, Investigation, Visualization, Methodology, Writing - original draft, Writing - review and editing; Dominique A Glauser, Conceptualization, Formal analysis, Supervision, Funding acquisition, Investigation, Visualization, Writing - original draft, Project administration, Writing - review and editing

## Author ORCIDs
Domenica Ippolito http://orcid.org/0000-0002-8793-5065
Dominique A Glauser http://orcid.org/0000-0002-3228-7304

## Decision letter and Author response
Decision letter https://doi.org/10.7554/eLife.85260.sa1
Author response https://doi.org/10.7554/eLife.85260.sa2

## Additional files

### Supplementary files
• Supplementary file 1. Plasmid name, cloning, and primer information.
• MDAR checklist

### Data availability
All data generated and analysed during this study are included in the manuscript and supporting file; Figure 1 - source data 1, Figure 2 - source data 1, Figure 3 - source data 1 and Figure 4 - source data 1 contain the numerical data used to generate the figures.

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
