## [Editor Report]

In this follow-up study to their previous work (Ippolito, 2021), the authors report additional insights into a complex network of nuclear export (NES) and nuclear localisation (NLS) sequences in the CaM kinase‐1 (CMK‐1), which is implicated in the plastic regulation of the FLP thermo‐nociceptors neurons.

---

## [Decision Letter]

**Decision letter after peer review:**

Thank you for submitting your article "Multiple antagonist calcium-dependent mechanisms control CaM Kinase-1 subcellular localization in a *C. elegans* thermal nociceptor" for consideration by *eLife*. Your article has been reviewed by 2 peer reviewers, and the evaluation has been overseen by a Reviewing Editor and Richard Aldrich as the Senior Editor. The following individual involved in the review of your submission has agreed to reveal their identity: Shawn Xu (Reviewer #2).

The reviewers have discussed their reviews with one another, and the Reviewing Editor has drafted this to help you prepare a revised submission. Overall, we are very excited about the work presented in this manuscript, but we think that a few revisions are needed to better support your major claims. We believe that the requested revisions are very feasible and can be addressed with some new experiments/data. Please address the points below and submit a point-by-point response alongside your revised manuscript.

Essential revisions:

*Reviewer #1:*

1) The authors identify a secondary NLS/NES structural element-pair that is, according to their claim, a basal, stimulation-independent structural modulator of CMK-1 subcellular distribution. The presented model suggests that this new NLS/NES pair regulates a basal shuttling rate of CMK-1 between the cytosol and the nucleus. However, this is not per se demonstrated in the current version of the manuscript. I suggest applying FLIP or iFRAP to demonstrate that these secondary elements are indeed regulating a basal nucleo-cytoplastic shuttling of CMK-1 in FLP neurons.

2) It is not clearly described how the authors ended up studying S325. Is this the only strongly predicted phosphorylation site in the whole CT part of the protein? Did the Authors specifically perform phosphorylation-site prediction to define candidate sites?

3) The axis of supplementary Figure 4 is clipped and some parts of the violin plots are missing.

*Reviewer #2:*

1) Figures 1-3 share some of the same control datasets. For example, it seems like the nucleus/cytoplasm ratios of CMK-1 wild-type animals at 20{degree sign}C and 28{degree sign}C are from the same trial of experiments in Figure 1B and 2C. If data are re-used in different figures, this should be clearly clarified and stated in figure legends.

2) Based on theory or data, it would be useful for the authors to be more specific about the extent to which a prolonged temperature rise and mutations in functional domains of CMK-1 would be expected to alter worms' FLP-related behaviors like heat avoidance reversals and heat adaptation. It would be helpful if the authors could include at least one functional behavioral assay to validate their findings in terms of the signaling pathway.

3) Line 106: In silico analysis tools first appearing in the manuscript should be briefly explained by a few sentences or a citation. Besides, a detailed explanation of this method is missed in the "Materials and methods" section.

4) Typos. Some examples:

Line 62: "through which" instead of "though which".

Line 65: "decrease CMK-1 nuclear accumulation".

Line 194/ 218/ 236: in the three sub-headings "T179D phosphor-mimic mutation" could be replaced by "T179 phosphorylation".

---

## [Author Response]

Reviewer #1:1) The authors identify a secondary NLS/NES structural element-pair that is, according to their claim, a basal, stimulation-independent structural modulator of CMK-1 subcellular distribution. The presented model suggests that this new NLS/NES pair regulates a basal shuttling rate of CMK-1 between the cytosol and the nucleus. However, this is not per se demonstrated in the current version of the manuscript. I suggest applying FLIP or iFRAP to demonstrate that these secondary elements are indeed regulating a basal nucleo-cytoplastic shuttling of CMK-1 in FLP neurons.

We agree that it would be very nice to be able to expand the analysis of our model with quantitative kinetic data. We have actually invested a lot of time and energy (and money) in attempting to establish in vivo FRAP in our lab over the last year, but we must admit we reached our limits. There are a lot of technical difficulties associated with the need to work with a moving whole animal preparation (and not immobile cells as used in the vast majority of FRAP studies). We tried to establish a micro-fluidic system, in order to avoid the need for interfering pharmacological agents, but we were never able to obtain sufficient stability of the head region. Furthermore, to obtain conclusive data in these tiny worm neurons with very limited signal area and relatively weak signal-to noise ratio, we would need to photobleach an entire compartment and measure its ‘replenishment’ kinetics over very long periods. But we were not so far able to achieve this with our current resources. Therefore, in the scope of the present revision, the suggested experiments are unfortunately out of our reach. To nevertheless address this concern, we revised the Result/Discussion to emphasize the need for quantitative kinetics analyses in future studies. The relevant text reads as follows:

“Our studies have identified multiple functional intrinsic elements, including NES, NLS and phosphorylation sites, and provide a more complete picture of their interplay in FLP thermo-nociceptor neurons. These elements have antagonistic effects, favoring either export or import, and in the case of the T179 phosphorylation, favoring both export and import. We speculate that the dual regulatory role of T179 phosphorylation in promoting bidirectional CMK-1 translocation across the nuclear envelope could be at the origin of cell stimulation-dependent CMK-1 shuttling cycles. Importin/exportin based transport is an energy consuming process. Therefore, if a single phosphorylation event promotes the two antagonistic pathways, it must also come with some benefits. Considering the cytoplasmic localization of CKK-1, we propose that a dynamic nucleo-cytoplasmic shuttling might contribute to ensure that the T179 phosphorylation status of CMK-1 is frequently “refreshed”, in order for the nuclear pool of CMK-1 to reflect the current activation status of CKK-1 in the cytoplasm. In order to test this hypothetical model, future studies will need to quantify the actual shuttling kinetics of wild type and mutant CMK-1 at different temperature. Furthermore, the whole process is likely to be also influenced by the activity of phosphatase(s) responsible for CMK-1 dephosphorylation on T179. In order to obtain a full picture of the phosphorylation-dependent CMK-1 localization regulation, future studies will also need to identify these phosphatases and determine their subcellular locus of action in vivo.”

We also carefully scanned our manuscript to make sure we were very cautious in our statements about the kinetic aspect and the baseline shuttling aspect. We made some text adjustments in the abstract and in the Result/Discussion section.

While the question of the shuttling kinetics will unfortunately remain open, we believe that our manuscript now fairly addresses this aspect without overselling our speculations about the potential role of activating two opposite drives. In addition, we want to stress that, the absence of shuttling kinetic data do not affect our main conclusions about the multiplicity of the antagonist regulatory mechanisms controlling CMK-1.

2) It is not clearly described how the authors ended up studying S325. Is this the only strongly predicted phosphorylation site in the whole CT part of the protein? Did the Authors specifically perform phosphorylation-site prediction to define candidate sites?

There are only 5 potential phosphosites in the considered C-terminal region of CMK-1. S325 and S327 received a high score (>0.7) using NetPhos-3.1. S327 prediction is not associated with any particular kinase, whereas S325 is a potential target of kinases known to regulate neuronal function (including PKA and CaMkinase themselves), and is part of the prevalent RXXS motif. We chose to follow up on S325 only for these reasons. We didn’t make any claim about the other residues.

In the revised manuscript, we have revised the text to cite the NetPhos-3.1 prediction tool and the very interesting article by Bradley and colleagues supporting the prevalence of the RXXS motifs (Bradley *et al.* 2019 10.1371/journal.pbio.3000341). These were the only criteria that drove our initial intention and ultimately, we were lucky to follow this fortunate intuition.

3) The axis of supplementary Figure 4 is clipped and some parts of the violin plots are missing.

We expanded the X-axis range in the revised supplementary figure 4 to make sure all the datapoints are visible.

Reviewer #2:1) Figures 1-3 share some of the same control datasets. For example, it seems like the nucleus/cytoplasm ratios of CMK-1 wild-type animals at 20{degree sign}C and 28{degree sign}C are from the same trial of experiments in Figure 1B and 2C. If data are re-used in different figures, this should be clearly clarified and stated in figure legends.

We complemented the legends of the relevant figures (2, 3) accordingly for more clarity.

2) Based on theory or data, it would be useful for the authors to be more specific about the extent to which a prolonged temperature rise and mutations in functional domains of CMK-1 would be expected to alter worms' FLP-related behaviors like heat avoidance reversals and heat adaptation. It would be helpful if the authors could include at least one functional behavioral assay to validate their findings in terms of the signaling pathway.

We agree that it is important that the present mechanistic follow-up study describes a physiologically relevant process. There is substantial evidence it is the case. Indeed, previously published data demonstrated that the subcellular localization of CMK-1 is essential to adjust FLP-dependent heat avoidance. In our two previous studies (Schild et al. 2014 and Ippolito et al. 2021), we examined a series of mutants and exogenous chimeric proteins (e.g., with appending of dominant canonical NLS or NES) and clearly showed that CMK-1 nuclear localization is necessary and sufficient to reduce heat avoidance, whereas CMK-1 cytoplasmic localization is necessary and sufficient to promote heat avoidance. Therefore, it is reasonable to expect that any new mechanisms that we show to be able to modulate CMK-1 localization is likely to be of potential physiological relevance. As for the new elements discovered here, it is unfortunately very difficult to design a clearly interpretable behavioral experiment that would add any further solidification on this aspect. First, regarding the new NES/NLS pair, it is a set of secondary elements, whose impact only becomes very salient when the primary NES/NLS pair is impaired. Performing behavioral experiments in a background combining three or more mutations would become tricky to interpret. Any result supporting our model would be a very weak support, whereas we could easily justify a result not fulfilling our expectations, because of a severe chronic miss-regulation of CMK-1. Second, regarding the T179 affecting mutations, they are even more challenging to work with. Indeed, while T179A and T179D produce major effects on CMK-1 localization on their own, they also create strong loss and gain of CMK-1 kinase activity, respectively. Regardless of the behavioral phenotypes we would obtain, we would not be able to disentangle the contribution of kinase activity versus localization.

To follow up on this comment, we modified the introduction to more explicitly refer to past experiments having demonstrated the functional importance of CMK-1 localization.

3) Line 106: In silico analysis tools first appearing in the manuscript should be briefly explained by a few sentences or a citation. Besides, a detailed explanation of this method is missed in the "Materials and methods" section.

This a very good point. In the revised manuscript, we have introduced the suggested information under a new specific Method section subheading, and also provide citation about these tools in the Results and Discussion section.

4) Typos. Some examples:Line 62: "through which" instead of "though which".Line 65: "decrease CMK-1 nuclear accumulation".Line 194/ 218/ 236: in the three sub-headings "T179D phosphor-mimic mutation" could be replaced by "T179 phosphorylation".

We corrected these typos and take another deep look at the manuscript to remove remaining typos.